# Optimization of Low-Tech Protected Structure and Irrigation Regime for Cucumber Production under Hot Arid Regions of India

**DOI:** 10.3390/plants13010146

**Published:** 2024-01-04

**Authors:** Pradeep Kumar, Pratapsingh S. Khapte, Akath Singh, Anurag Saxena

**Affiliations:** 1ICAR—Central Arid Zone Research Institute, Jodhpur 342003, India; 2ICAR—National Institute of Abiotic Stress Management, Baramati 413115, India; 3ICAR—Central Institute for Subtropical Horticulture, Rehmankhera, Lucknow 226101, India; akath.singh@icar.gov.in; 4ICAR—National Dairy Research Institute, Karnal 132001, India; anurag.saxena@icar.gov.in

**Keywords:** greenhouse cultivation, climate change, vegetables, productivity, sustainable production

## Abstract

Water scarcity and climate variability impede the realization of satisfactory vegetable yields in arid regions. It is imperative to delve into high-productivity and water-use-efficient protected cultivation systems for the sustained supply of vegetables in harsh arid climates. A strenuous effort was made to find suitable protected structures and levels of irrigation for greenhouse cucumber production in hot arid zones of India. In this endeavor, the effects of three low-tech passively ventilated protected structures, i.e., naturally ventilated polyhouse (NVP), insect-proof screenhouse (IPS) and shade screenhouse (SHS), as well as three levels of irrigation (100%, 80% and 60% of evapotranspiration, ET) were assessed for different morpho-physiological, yield and quality traits of the cucumber in a two-year study. Among the low-tech protected structures, NVP was found superior to IPS and SHS for cucumber performance, as evidenced by distinctly higher fruit yields (i.e., 31% and 121%, respectively) arising as a result of higher fruit number/plants and mean fruit weights under NVP. The fruit yield decreased in response to the degree of water shortage in deficit irrigation across all protected structures. However, the interaction effect of the protected structure and irrigation regime reveals that plants grown under moderate deficit (MD, 20% deficit) inside NVP could provide higher yields than those obtained under well-watered (WW, 100% of ET) conditions inside IPS or SHS. Plant growth indices such as vine length, node number/plant, and shoot dry mass were also measured higher under NVP. The greater performance of cucumber under NVP was attributed to a better plant physiological status (i.e., higher photosystem II efficiency, leaf relative water content and lower leaf water potential). The water deficit increased water productivity progressively with its severity; it remained higher in NVP, as reflected by 20% and 94% higher water productivity than those recorded in IPS and SHS, respectively, across different irrigation levels. With the exception of total soluble solids and fruit dry matter content (which were recorded higher), fruit quality parameters were reduced under water deficit conditions. The findings of this study emphasize the importance of considering suitable low-tech protected structures (i.e., NVP) and irrigation levels (i.e., normal rates for higher yields and moderate deficit (−20%) for satisfactory yields) for cucumber in hot arid regions. The results provide valuable insights for growers as well as researchers aiming to increase vegetable production under harsh climates and the water-limiting conditions of arid regions.

## 1. Introduction

Agriculture in the world’s hot arid climates faces formidable challenge owing to harsh climatic and edaphic factors [1]. Indian hot arid regions in particular are characterized by low soil fertility, low and erratic rainfalls, and high solar radiation and wind velocity (driving high rates of evapotranspiration) [2,3]. The relatively low crop productivity and high water requirements for a given quantity of produce are two major challenges in arid agriculture, so any effort directed towards increasing crop productivity should take into account the sustainable management of this precious resource [4,5,6]. Hence, the conservative use of water becomes a top priority in arid zone production systems, to meet the food and nutritional demands of habitants residing in these regions [7].

Vegetables constitute a crucial part of a nutritious diet; hence, the demand for fresh vegetables is increasing in every household. Vegetables are relatively highly resource responsive, requiring adequate amounts of water and nutrients with a favorable environment for optimum production [8]. However, their production in arid climates is more challenging, and it is difficult to obtain satisfactory yields of quality produce. This necessitates the development of region-specific highly productive and resource-use-efficient practices for sustainable vegetable production in resource-scarce hot arid ecosystems. One such approach is the adoption of protected cultivation technology, which can serve as a sustainable and viable solution to meet the escalating nutritious food demands of the growing population [9,10].

Due to its multifaceted benefits, the use of protected cultivation is rising across diverse regions around the world, ranging from high-altitude temperate climates to tropical and sub-tropical regions, Mediterranean regions, humid and sub-humid regions, and hot-arid areas, using suitable greenhouse structures [11,12,13]. The protected structures range from state-of-the-art energy-intensive climate-controlled greenhouses to low-tech structures such as passively (naturally) ventilated greenhouses and screenhouses that demand minimal energy input [11,14]. The cultivation of crops in these protected structures offers several advantages, including protection of crops from direct damage from erratic weather and high irradiance. Such structures have the ability to regulate microclimatic conditions, fostering better plant growth and development, extending the period available for crop harvesting [15], and even making crop cultivation possible out of the normal season [13]. However, it is important to note that unlike hi-tech climate-controlled greenhouses, there is a limited control over microenvironmental modifications within low-tech protected structures [11], so external environmental factors may influence the microenvironment and subsequently the crop productivity [13].

In passively ventilated greenhouses such as naturally ventilated polyhouse and screen or net houses, covering (shade) and ventilation are the main attributors of microenvironment modification [11]. The greenhouse cladding, because of its specific characteristics, has a notable impact on the alteration of the greenhouse microenvironment [11,16,17]. Greenhouse covers alter both the quantity and quality of radiation passing through them, depending on their type, color and shading factors [12,18]. Furthermore, the optical properties of the covering material also play a crucial role in influencing air temperature and humidity within the protected structures. These changes subsequently affect the photosynthetic efficiency [19] and respiratory needs of crops, potentially leading to alterations in water productivity [20,21]. It has been found that the modifications in light transmission, temperature, and vapor pressure deficits were more favorable in double-layer polyethylene-covered structures compared to those covered with UV-stabilized polyethylene, IR absorbers polyethylene, or normal polyethylene [22]. Hence, it is evident that the cladding material can influence the microenvironment of particularly passively controlled protected structures; the changes may vary with the agro-ecological regions. Particularly in arid regions, protected structures can be more useful in increasing vegetable production through mitigating the adverse effects of erratic weather and abiotic stresses, as well as in the optimized use of irrigation water under protective cover. In view of this, it is imperative to find the optimum protected structure coupled with irrigation levels for a highly popular cucumber crop in arid regions of India. 

Cucumber (*Cucumis sativus* L.) is one of the world’s most widely cultivated economic vegetable crops [23]. In India, seedless (parthenocarpic) or mini-cucumber is the most popular greenhouse vegetable, particularly under hot-arid regions, where open field cultivation is meager [10,24]. Cucumber is a fast-growing shallow-rooted crop and is sensitive to soil moisture deficits [25]. However, it is emphasized that a consistent shortage of irrigation water has to some level prompted a shift in irrigation management, from emphasizing production per unit area to maximizing production per unit of water consumed [26]. Optimization of irrigation is a critical component in vegetable production in arid regions and enhances efficiency for the water applied [23]. It involves the controlled application of water-based crop water demand. Proper irrigation management is essential for ensuring adequate moisture levels in the soil, promoting healthy plant growth, and maximizing crop yield. In conditions of water deficit beyond the tolerance limit, the rate of leaf expansion can reduce, leading to a reduction in leaf area and ultimately resulting in a decrease in the net photosynthetic area per plant [27]. Consequently, when plants are subjected to sub-optimal irrigation levels, both the net photosynthetic area and rate may decline, reflected in the reduction of overall photosynthesis [28]. This reduction in overall photosynthetic efficiency may negatively affect plant growth, resulting in yield losses under water-stress conditions. Water stress can have a severe impact on yield and quality, particularly in vegetable crops such as cucumber, which is highly vulnerable to water stress [29]. The quantity of irrigation water has a substantial influence on cucumber yields across all growth stages. Irrigation levels experiencing water deficiencies during the fruit development stages were found to be least productive, as documented by Mao et al. [30].

It was hypothesized that the use of suitable passively ventilated protected structures in conjunction with optimal irrigation levels, by ensuring high yields with optimized water use, will represent a sustainable means of cucumber production in hot arid regions. To achieve this, the performance of greenhouse cucumber was studied in three protected structures and under three irrigation levels, by analyzing various important morphological, physiological and agronomical traits in two growing seasons.

## 2. Materials and Methods

### 2.1. Protected Structures and Growing Conditions

The present research was carried out in the Precision Farming Block of the ICAR-Central Arid Zone Research Institute, Jodhpur (26°15′ N latitude, 72°59′ E longitude) from August to November in 2018 and 2019. In this study, three passively (naturally) ventilated low-tech protected cultivation structures of similar floor area (128 m^2^) were used: a naturally ventilated polyhouse (NVP), an insect-proof screenhouse (IPS), and a shading net house (SHS). All the structures were dome-shaped, of 8 m width × 16 m length with a 2.5 m side height and 4 m ridge height, composed of a structural frame of tubular galvanized iron of 2 mm thickness. The NVP was cladded with 200 µ-thick 5-layer UV-stabilized polyethylene, with fixed top arches but rollable side walls, beneath which a 40-mesh insect-proof screen was fixed. There was a 1 m top vent in the NVP and a 2 m rollable side vent on all sides. For the IPS, the same screen was used to cover all sides and top arches. A manually foldable white shade net (50% shading) was placed over trellis wire in both the NVP and IPS, whereas the SHS was cladded with a green shade screen (50% shading) from all sides and top arches of the dome, without provision of a shading net. The experiment was laid in a two-factorial split plot design, where structures were considered as the main plot and irrigation levels were the sub-plots.

The climatic conditions of the study area are hot and arid, characterized by significant diurnal and seasonal temperature fluctuations, low humidity, high solar radiation and high wind speeds, and a high rate of evapotranspiration [2,3]. The microclimatic data recorded inside the protected structures during crop growth are summarized in Table 1. The presented data are the monthly averages of each week, recorded on a clear day from morning till evening at hourly intervals. Air temperature and relative humidity were monitored using an Assmann psychrometer (Model MR-58, Hisamatsu, Tokyo, Japan), while photosynthetically active radiation (PAR) was measured with a line quantum sensor (MQ-301, Series#1178, Apogee, Logan, UT, USA).

The soil of the experimental site had the following basic characteristics: organic carbon content, 0.22%; pH, 7.8; and total nitrogen (N), available phosphorus (P) and potassium (K) contents, 0.03%, 16.3 kg ha^−1^ and 221.5 kg ha^−1^, respectively. The soil consisted of 85% sand, 8.1% silt and 5.5% clay. According to the US soil taxonomy, this soil is classified as coarse-loamy, mixed, hyperthermic Camborthid. The quality of ground water used for irrigation was pH 7.8 and EC 1.5 dS/m.

Prior to bed preparation for the planting of cucumber seedlings, 25 t ha^−1^ of farmyard manure and 1.0 t ha^−1^ of neem cake were mixed into the topsoil. The seedlings of commercially cultivated parthenocarpic and gynoecious cucumber hybrid Terminator (Yuksel Tohum Seeds, Himmatnagar, India) were prepared in 32 mm-cell plug-trays filled with a soilless medium (vermiculite: cocopeat; 1:2 ratio *v*/*v*) under the climate-controlled greenhouse. Fifteen-day-old seedlings were transplanted to paired rows at a spacing of 50 cm × 40 cm (3.0 plants m^−2^) on soil beds (9 inches height) in all the structures. Single main leader stems were maintained by regularly removing axillary shoots. Vines were supported with plastic twine attached to overhead trellis wires. During the cropping period, a ratio of 200:200:350 kg ha^−1^ for N:P:K was used, along with 50 kg of calcium (Ca) and 30 kg of magnesium (Mg) applied through daily fertigation using water-soluble fertilizers (Coromandel International Ltd., Secunderabad, India). Uniform crop management practices, including pruning, training, fertigation and crop protection were consistently implemented across all the structures. 

### 2.2. Irrigation Treatment and Water Productivity

The daily water requirement for cucumber crops was estimated using crop evapotranspiration (ET), which accounts for the volume of water transpired by the crop plus daily water losses occurred through evaporation from the soil-surface. The calculation of water volume for daily application through drip irrigation (in liters) was determined by the procedure described by Meshram et al. [31]:$$WR=\frac{\left(E_{pan}\times K_{p}\times K_{c}\right)}{ŋ}$$
where WR is the daily water requirement (L), E_pan_ is the daily pan evaporation (mm), K_p_ is the pan coefficient, K_c_ is the crop factor and η is the irrigation efficiency.

The daily pan evaporation reading (mm) was obtained from a US Class A open panevaporimenter, located about 100 m away from the protected structures. The crop factors were 0.6, 1.15 and 0.75 for initial (50 days), mid (50 days) and end (20 days) growth stages of the cucumber. The pan coefficient and irrigation efficiency were considered as 0.85 and 95%, respectively, for estimating the ET-based daily water requirement.

Furthermore, the time of daily drip irrigation was calculated by the following formula:$$T=\frac{WR}{Q\times n}$$
where T is the irrigation time (min. day^−1^), WR is the daily water requirement, Q is the discharge rate of emitter (LPH), and n is the number of emitters per m^2^

The three irrigation treatments were 100%, 80% and 60% of ET (or WR), considered as normal or well-watered (WW), moderate deficit (MD) and severe deficit (SD), respectively; these were uniformly applied across all protected structures. The moisture content was tested at two occasions (30 d and 55 d) using a time-domain reflectometry moisture meter in the top-soil (10 cm depth); it was 84.2–86.8% of field capacity (13.8%, *v*/*v*), indicating that the volume of water applied in normal irrigation (i.e., 100% of ET) was in an adequate range to maintain proper soil moisture. A precise irrigation delivery was assured by using 16 mm inline drip laterals with an emitter capacity of 1 L per hour and a 30 cm spacing between emitters, applied daily in the morning hours (Table 2). The uniformity in the emitter discharge was assured by regular monitoring and cleaning. 

Water productivity (WP, kg per m^−3^) was calculated following the method of Dermitas and Ayas [32] and expressed as the ratio of total yields to total irrigation water applied, during the entire growing period.

### 2.3. Plant Growth and Fruit Yield

Plant growth and yield-related parameters were assessed on five plants arbitrarily selected and tagged in each treatment within each replication. At the end of the experiment, leaves and stems were separated, dried in an oven at 65 °C until a constant weight was achieved, and measured for their dry weight (DW). Leaf area was measured using a leaf area meter (LI-3100C, LI-COR, Inc., Lincoln, NE, USA). During each harvest, the fruit number and weight were recorded for each tagged plant in all treatments; by aggregating these values across all harvests per plant, the total fruit number and yield were determined. The mean fruit weight was calculated by dividing the total fruit weight by the fruit number.

### 2.4. Fruit Quality

To assess fruit quality, samples of 15 uniform fruits were selected from bulk harvest from each replication. Fruits samples were taken for analysis at the peak of harvesting (55 days after transplanting). The following fruit parameters were measured in the laboratory: fruit length and girth (cm), fruit firmness (kg cm^−2^), total soluble solids (TSS, °Brix), and fruit dry matter (DM, %). Fruit TSS content was determined using a digital handheld refractometer (Bellingham and Stanley, Tunbridge Wells, UK). Fruit firmness was measured with a digital fruit-hardness tester (model no FR-5120, Lutron Electronic Enterprise Co., Ltd., Taipei, Taiwan) by puncturing the fruits with the plunger (6 mm diameter and 15 mm long) at two opposite positions and recording the pressure required (kg cm^−2^). Fruit DM content was determined by weighing dried fruit, as obtained by drying in an oven at 70 °C until a constant weight was reached.

### 2.5. Physiological Parameters

Physiological indices were recorded during the active growth period (50 days after transplanting). Total chlorophyll content was determined in fresh leaf tissues, and the total chlorophyll concentration was measured following a method suggested by Arnon [33]. Pigments were extracted using 80% aqueous acetone and estimated by measuring absorbance at 645 nm and 663 nm on a spectrophotometer, with the concentration expressed in µg mL^−1^ fresh weight. Leaf water potential was measured with a pressure chamber (Model 600, PMS Instrument Co., Corvallis, OR, USA). Leaf relative water content (RWC) was determined as per the method described by Khare et al. [34]. Twelve leaf discs for each measurement were weighed to determine fresh weight (FM) and rehydrated in distilled water for 6 h; then, the turgid leaf discs were surface-dried and weighed again to obtain the turgid weight (TM). Subsequently, the same discs were oven-dried at 80 °C for 24 h to determine dry weight (DM). The PS II efficiency (Fv/Fm) was measured on the 3rd–4th fully opened leaves using a chlorophyll fluorescence meter (OS-30p, Opti-Sciences, Inc., Hudson, NH, USA).

### 2.6. Statistical Analysis

Statistical analysis of the collected research data was conducted using the R statistical software, version 4.3.2 (R Core Team, 2021). Before proceeding with the statistical analysis, the data underwent a normality check using the Shapiro–Wilk normality test. To analyze two factors, a split-plot design was employed, utilizing the Doebioresearch and agricolae packages in R. For post-hoc mean comparisons, the least-significant difference (LSD) test was applied. Additionally, the statistical plots were generated using GraphPad Prism (version 10.1.0).

## 3. Results 

### 3.1. Morphometric Traits

Protected structures and irrigation levels significantly affected different morphometric traits of the cucumber plants (Table 3). The interaction effect of protected structure (main factor) and irrigation regime (sub-factor) was noticed only on the leaf area, node number and stem girth determined with pooled data for the two years. As a main factor effect, the naturally ventilated polyhouse (NVP) outperformed the screenhouses (IPS and SHS) for various growth metrics, regardless of whether the plants were grown under well-watered (WW) conditions or water deficit conditions (moderate deficit, MD, and severe deficit, SD). The plants grown inside NVP displayed distinctly higher mean shoot dry masses (12% and 13% higher) and leaf areas (13% and 15% higher) than IPS and SHS, respectively; however, the two screenhouses did not show statistical variation from each other for these analyzed traits. The sub-factor irrigation level had more conspicuous effects on both shoot dry mass and leaf area. These traits tended to decrease with increasing water deficit levels: the shoot dry mass was reduced by 19% and 40% and the leaf area reduced by 18% and 28% under MD and SD conditions as compared to the WW condition (Table 3). Meanwhile, the interaction effect of the protected structure and irrigation level indicates that the leaf area measured under MD inside NVP was similar to that measured under WW conditions inside IPS (Figure 1). Vine length was also recorded higher in the NVP compared to the screenhouses (IPS and SHS), and among irrigation levels, it was distinctly higher in the WW condition, 7% and 17% higher than the MD and SD conditions. Likewise, the number of nodes per vine was also recorded highest inside NVP and under WW conditions among the different protected structures and irrigation levels, respectively. The mean stem girth was at par under NVP and IPS, registering distinctly thicker stems than those recorded in SHS. Irrigation at normal rate (WW) produced robust plants with significantly higher stem girths across all the structures. Stem girth decreased with water deficit; however, it was almost similar between MD and SD in NVP and IPS, significantly lower in SHS (Figure 1). Overall, the NVP in conjunction with normal irrigation (WW) was found the most optimum combination to produce much healthier and stronger cucumber vines.

### 3.2. Fruit Quality Attributes

The fruit’s physical as well as internal quality attributes were affected by both the main factor (protected structures) and sub-factor (irrigation levels) (Table 4). Among the protected structures, the fruit’s physical attributes (such as fruit length and fruit girth) were recorded highest in NVP supplied with normal irrigation, both under independent and combined conditions of these two factors (Table 4). The mean fruit length was 5% and 6% higher in NVP compared to the IPS and SHS, respectively, and it was 7% and 13% higher in WW compared to MD and SD conditions, respectively. Interaction data in Figure 2 revealed that fruit length was apparently higher under NVP combined with WW, and it slightly decreased under deficit irrigation, especially under SD conditions, though it had statistically similar values between IPS and SHS under both MD and SD conditions. In cucumber, the fruit quality parameters such as firmness, TSS and fruit dry matter were significantly affected by irrigation levels; however, among these, only TSS was affected by growing conditions in different protected structures. These quality parameters were recorded distinctly higher in SD followed by MD and WW conditions, regardless of protected structures. Furthermore, TSS was significantly higher in SHS than NVP or IPS over irrigation levels. The interaction effect of structure and irrigation level showed that TSS and fruit dry matter content were statistically similar between NVP and IPS when grown under WW or MD; however, using SD, their contents were changed significantly, suggesting that cucumber fruits produced under low water-supply conditions produced greater soluble solids and fruit dry matter as compared to the MD and (especially) WW conditions (Figure 2).

### 3.3. Physiological Parameters

Maintaining an optimum plant water balance under areas of harsh climate with water scarcity is of paramount importance to achieving high yields. However, this balance is regulated by multiple production variables. Among these, the microenvironment of the protected structures and water availability in the rhizosphere are of prime importance. Based on the two years’ pooled data, it is clear that with the exception of chlorophyll, no physiological parameters were influenced by the main factor (i.e., growing environment) in this study. This leaf pigment was higher in NVP (10% and 26%) than IPS and SHS, respectively (Table 5). However, the mean effect of sub-factor irrigation level was much more pronounced over that of protected structure: the chlorophyll content increased with an increase in water deficit, but a significant increase was evident only under SD conditions. The photosystem II (PS II) is considered one of the most reliable indices to judge a plant’s physiological efficiency, particularly under water deficit conditions; its efficiency decreased under an increase in water deficit, but the effect was pronounced only under SD conditions. As expected, the leaf water potential (WP) increased under an increase of water deficit levels. Accordingly, WP was significantly higher under SD followed by MD and WW. In contrast to WP, the leaf relative water content (RWC) tended to decrease progressively under an increase in water deficit. It can be inferred that the quantum of variation was greater with regard to the sub-factor water supply than the main-factor protected structure for studied physiological indices for cucumber production in arid conditions.

### 3.4. Yield Parameters and Water Productivity

Both protected cultivation structures and irrigation levels caused significant effects on yield-attributing traits, independently of each other (Table 6). The fruit number under NVP was 22% and 81% more than under IPS and SHS, respectively. Meanwhile, in the case of irrigation levels, it was 9% and 23% higher under WW conditions over MD and SD ones. Likewise, fruit weight exhibited a similar trend to that of fruit number for structures as well as irrigation levels. Fruit yield varied among the structures and irrigation levels, as it is a highly dependent trait. NVP outperformed the screenhouses. In NVP, the mean cucumber fruit yield over all irrigation levels was 30% and 121% higher than those recorded inside IPS and SHS, respectively. Since irrigation is an important factor in protected cultivation, its shortage supply can clearly impact crop yield. Fruit yield decreased under an increase in water deficit severity across all protected structures; however, the highest yield was obtained in the WW condition, recording a 17% and 43% higher yield over MD and SD conditions, respectively. Interestingly, the interactive effects of structure and irrigation level showed that cucumber grown under MD inside NVP registered slightly higher yields than that obtained under WW inside IPS, indicating a significant water saving could be possible with the use of NVP as compared to screenhouses (Figure 3). Conversely, water productivity tended to increase with the degree of reduction in water supply across all protected structures. However, the water productivity was apparently higher under NVP (i.e., 94% and 20%) as compared to SHS and IPS, respectively, across the irrigation levels (Table 6).

## 4. Discussion

Considering the increasing demand for fresh vegetables, production-related challenges in harsh arid climates, and scarce water availability, there is a need to devise a climate-risk-proof and water-use-efficient growing system. Hence, this study projected to find a suitable protective cultivation structure and irrigation regime for growing (mini)cucumber, a highly popular greenhouse vegetable, by assessing cucumber performance in three passively ventilated low-tech protected structures coupled with three irrigation levels in two consecutive years (2018 and 2019).

Protected cultivation structures and irrigation regimes significantly influenced various aspects of cucumber production in an independent or integrative manner. The most favorable response of a protected structure on cucumber vine growth was under NVP, reflected by the higher shoot dry mass, leaf area, vine length and node number per vine as compared to those observed under screenhouses (IPS and SHS). While comparing the two screenhouses, most of the plant growth indices were recorded higher under IPS compared to SHS, except for vine length (Table 3). The better plant growth in NVP seemed to be due to the more congenial microenvironment observed inside plastic-covered structures than screen/net-covered structures in IPS or SHS. The pronounced growth observed in greenhouse cucumber plants [35] and tomato plants [19] in previous studies was associated with a positively altered microclimate inside plastic-covered greenhouses. In net or screenhouses, only regulation in the amount of direct solar radiation falling onto the plants was possible, thereby causing shading effects which possibly hampered the plants’ physiological processes. Meanwhile, in the case of a plastic-covered NVP, besides the possible changes in light quantity, the diffusion of light due to the inherent properties of plastic cover and the relatively high retention of air moisture inside plastic (particularly in hotter periods) might cause favorable effects on plant physiological processes, which are reflected in plants growth [11,36,37].

It is stated that to attain optimum plant growth, an optimum balance between the crop water demand and enhanced metabolism of plants is needed [5]. In the present study, the supply of daily irrigation at normal rate, as determined by the standard crop evapotranspiration (WW, 100% ET) through drip could help achieve distinctly higher plant growth attributes, since these were reduced under moderate deficit (MD, −20% of ET) and (especially) severe deficit (SD, −40% of ET) conditions across all three protected structures. Although deficit irrigation has been widely recognized as an efficient tool to optimize water use in open-field cultivation [38] and fruit orchards [39], its advantage seems to be limited to protected cultivation, as evidenced from a significant reduction in different important growth attributes (e.g., leaf area, vine length, node number and shoot dry mass) under both moderate and severe water deficit conditions (Table 4). A previous study on tomatoes grown under different protected environments also reports a significant reduction in shoot growth as well as fruit yield parameters under sub-optimal water supply conditions [19]. Furthermore, arid environments are characterized by not only limited precipitation but also by high evapotranspiration, which results in the accumulation of salts in the uppermost soil layer. Improved cucumber growth in the WW condition may be linked to the transport of salts toward the outer periphery of the soil’s wetting front. Conversely, in the context of deficit irrigation levels, this may lead to a reduction in soil osmotic potential, inducing osmotic stress in cucumber plants [5]. Furthermore, water deprivation might cause osmotic imbalances and damage to cellular components, thereby resulting in the inhibition of shoot growth, more conspicuously so in severe water deficit conditions [40]. This is quite evident when observing a severe reduction in shoot dry mass, leaf area and vine length under SD in comparison with MD, regardless of the protected structures. In fact, under protected cultivation, vegetables are accustomed to a regular supply of water [41]; hence, plants receiving sub-optimal quantities of water under MD and SD could not meet their water demands to balance the simultaneous occurrence of vegetative and fruit growth [19]. This is why the conspicuous damaging effect of water shortage was obvious on various growth parameters, which was also reflected later on in the fruit yields.

The combination of a favorable microenvironment and adequate water supply to root zones in protected conditions has a direct influence on plant physiological processes and a subsequent effect on growth and yield parameters [19]. In the present study, both protected structures and irrigation levels significantly influenced the physiological status of cucumber plants. In NVP, plants were able to maintain higher photosystem II efficiency in combination with higher leaf chlorophyll and relative water content (RWC), along with relatively lower values of leaf water potential compared to screenhouses (IPS and SHS). In contrast, the lower values of various physiological as well as growth indies under water deficit (MD and SD) conditions were clearly evident due to the sub-optimal water status of plants (exhibited by lower-leaf RWC) and its associated effects on plant PS II efficiency. The chlorophyll content increased under water deficit conditions, most likely as a concentration effect under low water supply and an adjustment by reduction in leaf size and the concentration of leaf pigment. Earlier reports also highlight that a slight decrease in water deficit can enhance the chlorophyll content in tomatos [42] and cucumbers [43]. In NVP, optimal microclimatic variables have helped cucumber plants to perform better due to higher photosynthesis and less damage to the cellular membrane under water deficit conditions. Moreover, in NVP, the CO_2_ emitted by plants gets trapped during the night due to the non-porous cladding materials (polyethylene sheet), in contrast to net houses; this perhaps contributes to boosting of the photosynthesis efficiency. Water levels had a significant influence on the physiological traits of cucumbers. As the level of water stress deficit increased, there was a significant reduction in photosystem II efficiency, and plants required maximum energy to extract water from the soil profile, which is correllated with a higher leaf water potential. De Swaef [44] highlighted that it is important to understand the concept of plant water potential, in order to know how water moves through plants and their immediate environment. Since irrigation was applied daily in the morning, the cucumber plants were less stressed under the WW condition, which is reflected in the relatively lower leaf water potential, most likely due to lesser deviations in soil moisture from the field capacity as compared to those occurring under water deficit conditions. Microenvironment and soil water availability affected the leaf water potential in cucumber; it was lowest in NVP under WW irrigation levels, and under water deficit the values were higher, which indicates that plants were stressed. The overall cucumber plant physiological status was consistently better in NVP due to a better microenvironment, which was altered by the polyethylene cladding sheet; this helped to diffuse radiation and maintain better relative humidity levels [12], which aided cucumber plants grown in NVP to maintain a proper soil–plant–atmosphere continuum compared to net houses.

The fruit yield and yield-related attributes of cucumber plants were distinctly higher under NVP than the two screenhouses. The higher fruit number per plant in NVP was clearly associated with highest vine length and node numbers per vine. However, the greatest fruit weight—substantiated with higher fruit length and girth—as well as fruit yield could be attributed to a distinctly better plant-water balance, leading to better plant growth and physiological functioning of cucumber plants grown in NVP compared to those of screenhouses (IPS and SHS). Irrigation water levels had a considerable influence on yield-attributing traits: fruit yield, fruit number and fruit weight tended to decrease under an increase of water deficit. Particularly in the SD condition, it seems that cucumber plants were not able to obtain the water (and/or nutrients) needed for the developing fruits, due to simultaneous vegetative growth which might have hampered the source–sink balance, thereby resulting in reduced fruit size and yield. This was partly explained by the fact that a higher shoot DW and leaf area was associated with higher fruit yields, probably due to the better availability of food substrates to developing fruits under the WW condition, especially under the favorable conditions of NVP. In a previous work, the decrease in photosynthesis in cucumber plants was related to a reduction in leaf area and its subsequent effect on the decrease in fruit yield when exposed to water deficit [43]. The water deficit in growing medium leads to stomata closure and enhancement in canopy temperature, and this might affect photo-assimilate formation in cucumber leaves; a similar point was highlighted in greenhouse cucumber by Kaukoranta et al. [45]. Contrary to our results, Rahil and Qanadillo [46] reported the highest yield at moderate deficit (70% ET), which was similar to full ET irrigation levels in greenhouse cucumber. This disparity in results could be because they grew half the number of plants per square meter (i.e., 1.5 plants) compared to our study (i.e., 3.0 plants). This signifies sufficient soil moisture availability to sustain optimum growth and physiology, even under designated water deficit conditions, in addition to greenhouse design. Season and soil factors may also affect plant performances in the given situation [11]. In fact, the response of water supply on crop performance may vary with the design of protected cultivation structures [11,24]. In the present study, the interaction effect of types of protected structure and irrigation level reveals that the plants grown under MD inside NVP were more vigorous and productive compared with those grown under WW conditions under IPS or SHS (Figure 1 and Figure 3).

Fruit quality (physical and chemical) is an important consumer trait, particularly in a freshly eaten salad vegetable such as cucumber. The fruit growth defined by its size (length and girth) was clearly influenced by growing conditions. Similar to plant growth attributes, cucumber fruits harvested under NVP maintained better physical quality attributes (fruit length and girth) than those obtained from screenhouses, particularly shade screenhouses, which produced relatively low-quality produce (Table 4). Fruit length and girth were reduced under sub-optimal water supply (in MD and SD) in all protected structures, possibly due to disrupted source–sink balancing under water deficit conditions and its subsequent effects on developing fruits. Previous studies have also highlighted the reduction in fruit size of different greenhouse vegetables under limited water supply conditions, as well as the associated reduced plant growth [19,47]. The fruits’ internal quality attributes were least affected by protected structures, but the effects were clearly visible with respect to irrigation levels. Cucumber fruit contain more than 95% water; therefore, a reduction in fruit size may occur under water limiting conditions (MD and SD) in comparison with adequate water availability in WW conditions. Conversely, fruits obtained under water deficit conditions had better internal fruit quality parameters such as fruit firmness, TSS and fruit DM content. Several researchers have also reported improvements in certain biochemical parameters, including fruit soluble solids and dry matter contents in vegetable crops grown under abiotic stresses, due to stress-induced biosynthesis and/or due to the concentration effect [19,48].

The yield optimization with respect to available water quantity is a pre-requisite in water-scarce regions. Both protected structures and irrigation levels significantly affected the water productivity of cucumber plants. The assessment of water productivity for cucumber plants grown under respectively similar rates of water supply in different protected structures reveals that NVP proved to be better than IPS (followed by SHS) in terms of efficiency in water use. Interestingly, the water productivity of NVP was almost two-fold higher compared to SHS. The water productivity increased under an increase in water deficit across all structures. However, even though water productivity was highest under SD conditions, the quantum of yield level may not be promising, especially where the primary aim is yield maximization in high-investment protected cultivation. Many researchers have also highlighted that under water deprivation conditions, cucumber water productivity increases but overall yield decreases [34,49]. NVP, in combination with the WW level, was able to produce higher yields by modulating growth and maintaining better physiological status in arid regions.

## 5. Conclusions

It is clear from the present study that the naturally ventilated polyhouse was a more efficient protected structure for producing high yields and good-quality greenhouse cucumber fruits, with an optimum use of scarcely available water, as depicted by high water productivity among prevalent passively ventilated low-tech greenhouse structures. Furthermore, irrigation at below-normal rates was not found a feasible option of water saving for cucumber production in any of the protected cultivation structures, as evident from the significant yield reduction even under moderate levels of deficit irrigation (80% of ET). Hence, it is inferred that the combination of NVP with a normal rate of irrigation (100% of ET) can be considered most optimal for commercial cucumber cultivation in the water-scarce harsh arid climate of India or the world.

## Figures and Tables

**Figure 1 plants-13-00146-f001:**
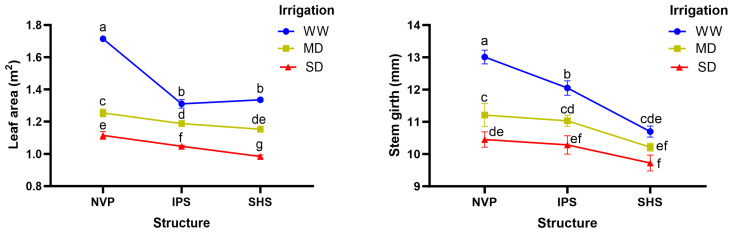
Interaction effect of protected structures and irrigation levels for leaf area (**left**) and stem girth (**right**); mean values followed by the same letter are not significantly different as computed by LSD (*p* ≤ 0.05).

**Figure 2 plants-13-00146-f002:**
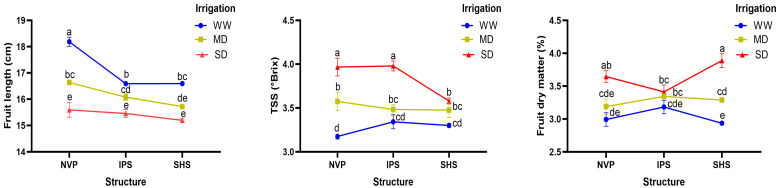
Interaction effect of protected structures and irrigation levels for fruit length (**left**), TSS (**center**) and fruit dry matter (**right**), showing the mean values followed by the same letter if not significantly different as per LSD (*p* ≤ 0.05).

**Figure 3 plants-13-00146-f003:**
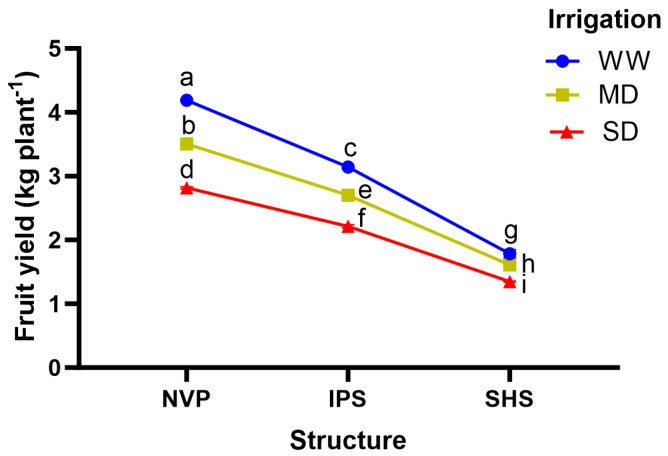
Interaction effect of protected structure and irrigation level for fruit yields, showing the mean values followed by the same letter if not significantly different as computed by LSD (*p* ≤ 0.05).

**Table 1 plants-13-00146-t001:** Mean monthly temperature (°C), relative humidity (RH, %) and photosynthetic active radiation (PAR, µmol m^−2^s^−1^) inside the protected structures during cucumber crop growth.

Years/Months	Temperature (°C)			RH (%)			PAR (µmol m^−2^s^−1^)		
	NVP	IPS	SHS	NVP	IPS	SHS	NVP	IPS	SHS
August, 2018	32.66	31.21	32.52	69.31	60.23	67.12	681	731	372
September, 2018	33.21	32.76	32.20	55.53	50.35	54.26	625	966	497
October, 2018	32.54	33.36	33.29	39.37	32.28	34.31	536	727	263
November, 2018	30.64	29.80	30.09	42.73	37.64	39.69	374	579	205
August, 2019	34.13	33.80	33.87	69.83	73.74	75.80	448	560	405
September, 2019	34.56	34.06	34.13	78.10	72.01	74.12	589	659	510
October, 2019	31.10	31.04	31.70	50.42	49.33	51.38	438	522	422
November, 2019	28.35	27.68	28.20	58.92	54.83	56.91	297	441	241

NVP: naturally ventilated polyhouse; IPS: insect-proof screenhouse; SHS: shade screenhouse.

**Table 2 plants-13-00146-t002:** Total amount of applied water in 2018 and 2019 during cucumber growth period in protected structures.

Irrigation Levels	Total Water Applied via Drip System (mm)	
	2018	2019
100% of ET (WW)	434.13	412.18
80% of ET (MD)	347.30	329.74
60% of ET (SD)	260.47	247.31

Note: We applied 60 mm water through a drip uniformly for all protected structures and irrigation treatments up to 21 days from transplanting for proper establishment of seedlings in both years.

**Table 3 plants-13-00146-t003:** Effect of protected structures and irrigation levels on different morphometric traits of cucumber plants.

Treatment	Shoot Dry Matter (g)			Leaf Area (m^−2^)			Vine Length (m)			Node Number			Stem Girth (mm)		
	2018	2019	Pooled	2018	2019	Pooled	2018	2019	Pooled	2018	2019	Pooled	2018	2019	Pooled
Structures (S)															
NVP	74.83 a	78.04 a	76.44 a	1.36 a	1.36 a	1.36 a	3.21 a	3.56 a	3.38 a	43.11 a	48.41 a	45.76 a	10.87 a	12.23	11.55 a
IPS	65.08 b	71.97 b	68.52 b	1.35 a	1.28 b	1.18 b	2.12 c	2.58 c	2.35 c	38.33 b	44.39 b	41.36 b	10.55 a	11.69	11.12 a
SHS	65.60 b	69.65 b	67.62 b	1.07 b	0.94 c	1.15 c	2.70 b	3.12 b	2.91 b	36.44 c	41.79 c	39.12 c	9.11 b	11.31	10.21 b
Irrigation (I)															
WW	80.88 a	85.57 a	83.22 a	1.54 a	1.36 a	1.45 a	2.89 a	3.30 a	3.10 a	41.89 a	47.25 a	44.56 a	11.40 a	12.43 a	11.91 a
MD	67.79 b	72.29 b	70.04 b	1.22 b	1.16 b	1.19 b	2.67 b	3.12 b	2.90 b	39.44 b	44.90 b	42.17 b	9.84 b	11.78 a	10.81 b
SD	56.84 c	61.79 c	59.32 c	1.02 c	1.07 c	1.04 c	2.46 c	2.84 c	2.65 c	36.56 c	42.45 c	39.50 c	9.28 c	11.02 b	10.15 c
S	***	**	***	***	***	***	***	***	***	***	***	***	**	NS	*
I	***	***	***	***	***	***	***	***	***	***	***	***	***	**	***
S × I	NS	NS	NS	***	***	***	NS	NS	NS	NS	NS	NS	***	NS	*

Mean values of three replicates followed by the same letter for each factor within each column if not significantly different according to LSD (*p* ≤ 0.05). NS, non-significant; significance *, ** and *** at *p* ≤ 0.05, 0.01 and 0.001, respectively. NVP, naturally ventilated polyhouse; IPS, insect-proof screenhouse; SHS, shade screenhouse; WW, well-watered (100% of ET); MD, moderate deficit (80% of ET); SD, severe deficit (60% of ET).

**Table 4 plants-13-00146-t004:** Effect of protected structures and irrigation levels on fruit quality traits of cucumber.

Treatment	Fruit Length (cm)			Fruit Girth (cm)			Fruit Firmness (Kg cm^−2^)			TSS (°Brix)			Fruit Dry Matter (%)		
	2018	2019	Pooled	2018	2019	Pooled	2018	2019	Pooled	2018	2019	Pooled	2018	2019	Pooled
Structures (S)															
NVP	16.82 a	16.78 a	16.80 a	4.26 a	4.32 a	4.29 a	3.93 a	3.74	3.84	3.54	3.60	3.57	3.24	3.31	3.27
IPS	15.82 b	16.25 ab	16.04 b	4.09 b	4.11 b	4.10 b	3.95 a	3.70	3.83	3.61	3.59	3.60	3.29	3.30	3.31
SHS	15.86 b	15.80 b	15.83 b	4.09 b	4.12 b	4.10 b	3.75 b	3.64	3.69	3.38	3.51	3.45	3.31	3.44	3.37
Irrigation (I)															
WW	16.92 a	17.31 a	17.12 a	4.29 a	4.32 a	4.30 a	3.45 c	3.43 c	3.44 c	3.20 c	3.34 c	3.27 c	2.96 c	3.10 b	3.03 c
MD	16.14 b	16.14 b	16.14 b	4.16 b	4.18 b	4.17 b	3.90 b	3.61 b	3.76 b	3.48 b	3.53 b	3.51 b	3.24 b	3.29 b	3.27 b
SD	15.44 c	15.38 b	15.41 c	4.00 c	4.04 c	4.02 c	4.28 a	4.03 a	4.16 a	3.85 a	3.82 a	3.84 a	3.63 a	3.66 a	3.64 a
S	***	*	**	*	*	*	*	NS	NS	NS	NS	NS	NS	NS	NS
I	***	***	***	***	***	***	***	***	***	***	***	***	***	*	***
S x I	**	*	**	NS	NS	NS	NS	NS	NS	NS	NS	*	***	NS	*

Mean values of three replicates followed by the same letter for each factor within each column if not significantly different according to LSD (*p* ≤ 0.05). NS, non-significant; significance *, ** and *** at *p* ≤ 0.05, 0.01 and 0.001, respectively. NVP, naturally ventilated polyhouse; IPS, insect-proof screenhouse; SHS, shade screenhouse; WW, well-watered (100% of ET); MD, moderate deficit (80% of ET); SD, severe deficit (60% of ET).

**Table 5 plants-13-00146-t005:** Effects of protected structures and irrigation levels on different physiological parameters of cucumber.

Treatment	Total Chlorophyll (µg mL^−1^)			PS II (Fv/Fm)			Ψ_leaf_ (-bar)			Relative Water Content (%)		
	2018	2019	Pooled	2018	2019	Pooled	2018	2019	Pooled	2018	2019	Pooled
Structures (S)												
NVP	14.11 a	15.18 a	14.65 a	0.70 a	0.83	0.77 a	12.44	6.07 b	9.26	67.95 a	68.23	68.09 a
IPS	10.56 b	12.74 b	11.65 c	0.65 c	0.82	0.75 b	12.66	6.41 a	9.52	65.23 b	67.94	66.58 b
SHS	13.45 a	13.13 b	13.29 b	0.68 b	0.82	0.74 b	12.77	6.32 a	9.56	63.69 b	68.34	66.02 b
Irrigation (I)												
WW	11.41 b	13.11	12.26 b	0.69	0.83 a	0.76 a	9.77 c	5.24 c	7.52 c	68.66 a	70.88 a	69.77 a
MD	12.34 b	13.63	12.98 ab	0.68	0.82 b	0.75 ab	13.11 b	6.32 b	9.71 b	65.48 ab	67.72 b	66.60 b
SD	14.38 a	14.32	14.35 a	0.68	0.81 c	0.74 b	15.00 a	7.24 a	11.12 a	62.73 b	65.91 b	64.32 c
S	**	*	**	***	NS	**	NS	**	NS	*	NS	*
I	**	NS	*	NS	***	*	***	***	***	**	**	***
S x I	NS	NS	NS	NS	**	NS	NS	NS	NS	NS	NS	NS

Mean values of three replicates followed by the same letter for each factor within each column if not significantly different according to LSD (*p* ≤ 0.05). NS, non-significant; significance *, ** and *** at *p* ≤ 0.05, 0.01 and 0.001, respectively. NVP, naturally ventilated polyhouse; IPS, insect-proof screenhouse; SHS, shade screenhouse; WW, well-watered (100% of ET); MD, moderate deficit (80 of ET); SD, severe deficit (60% of ET).

**Table 6 plants-13-00146-t006:** Effects of protected structure and irrigation level on yield attributes and water productivity of cucumber.

Treatment	Fruit Number (plant^−1^)			Fruit Weight (g)			Fruit Yield (kg plant^−1^)			Water Productivity (kg m^−3^)		
	2018	2019	Pooled	2018	2019	Pooled	2018	2019	Pooled	2018	2019	Pooled
Structures (S)												
NVP	22.38 a	24.83 a	23.61 a	153.28 a	135.59	144.43 a	3.43 a	3.57 a	3.50 a	33.14 a	36.55 a	34.85 a
IPS	17.88 b	20.77 b	19.33 b	139.36 b	132.14	135.75 a	2.49 b	2.88 b	2.68 b	28.48 b	29.50 b	28.99 b
SHS	12.16 c	13.88 c	13.01 c	121.84 c	122.37	122.10 b	1.48 c	1.67 c	1.58 c	17.02 c	18.87 c	17.94 c
Irrigation (I)												
WW	18.88 a	22.05 a	20.47 a	151.44 a	147.55 a	149.50 a	2.92 a	3.15 a	3.03 a	24.51 c	25.93 c	25.22 c
MD	18.22 a	19.50 b	18.86 b	133.91 b	123.85 b	128.88 b	2.48 b	2.72 b	2.60 b	26.13 b	28.00 b	27.07 b
SD	15.33 b	17.94 c	16.63 c	129.13 b	118.70 b	123.91 b	1.99 b	2.25 c	2.12 c	28.01 a	30.98 a	29.50 a
S	***	***	***	*	NS	**	***	***	***	***	***	***
I	**	***	***	**	***	***	***	***	***	***	***	***
S x I	NS	NS	NS	NS	NS	NS	***	***	***	NS	NS	NS

Mean values of three replicates followed by the same letter for each factor within each column if not significantly different according to LSD (*p* ≤ 0.05). NS, non-significant; significance *, ** and *** at *p* ≤ 0.05, 0.01 and 0.001, respectively. NVP, naturally ventilated polyhouse; IPS, insect-proof screenhouse; SHS, shade screenhouse; WW, well-watered (100% of ET); MD, moderate deficit (80 of ET); SD, severe deficit (60% of ET).

## Data Availability

Data are contained within the article.

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
