# Peer review of "Optimization of Low-Tech Protected Structure and Irrigation Regime for Cucumber Production under Hot Arid Regions of India"

_plants, 2024, doi:10.3390/plants13010146_

Round 1

Reviewer 1 Report

Comments and Suggestions for Authors

Review of the MS ID:  plants-2782539. Optimization of Low-tech Protected Structure and Irrigation Regime for Cucumber Production under Hot Arid Regions of India

Comments:

This work deals with the search for a suitable sheltered structure and irrigation system for greenhouse cucumber cultivation in hot, dry areas of India. In this two-year study, the passively ventilated, low-tech sheltered structures: naturally ventilated polyhouse, insect-proof screenhouse and shaded screenhouse were evaluated in combination with three irrigation regimes (100%, 80% and 60% of evapotranspiration, ET) for different morphophysiological, yield and quality traits of cucumber.

Although the results of the study could provide useful information on efficient protected structure combined with optimal utilisation of scarce water to produce high yields and high quality cucumber fruits, there are some points that should be addressed before MS is considered for publication in Plants.

Title: is OK

The introduction is well written! But how can the author determine the optimal amount of irrigation if he  has no sensors to monitor the soil water content?

M&M: The description of the experimental design is poorly written. It is a two-factorial experiment with a split-plot design. Please add information on which treatment was placed in the main plot and which in the subplot.

The statistical analysis should be corrected according to the experimental design. Please include a description of the statistical methods used to statistically analyse the data.

L 194: Which plunger was used to measure the firmness of the fruit??

Results: All results where the interaction of two factors was significant should be presented in a separate table or figure – as differences between the mean scores of the treatments with the standard errors.

P7, 8 and 10: The nature of the figures should be corrected. Histograms are not suitable for presenting the results of the averages of the treatments. Also, in this study, the results should be presented according to the experimental design: the main plots are structures and the subplots are irrigation regimes. Please correct all figures (1, 2 and 3).

L: 388: How did the authors achieve the normal water supply?

If the authors include screen and nets in their considerations, they should shown the restirction fo quanity and quality of irradiation within the structure!

When the authors talk about the different irrigation methods, they should show the dynamics of the water content in the soil. The main drawback of the manuscript is the lack of measurements of soil water content during the growing season. The morphological and physiological response to different microclimatic conditions under the different low-tech structures and different irrigation practices could be much more accurately represented if these types of measurements were included in the study.

Author Response

Dear Reviewer#1

We would like to thank you for providing valuable suggestions to improve the quality of manuscript. We have revised the manuscript considering the comments/ suggestion made by your goodself as well as other reviewers. The revision are highlighted in the Red font in the revised MS, and responses to your comments are provided below:

Comment #1: This work deals with the search for a suitable sheltered structure and irrigation system for greenhouse cucumber cultivation in hot, dry areas of India. In this two-year study, the passively ventilated, low-tech sheltered structures: naturally ventilated polyhouse, insect-proof screenhouse and shaded screenhouse were evaluated in combination with three irrigation regimes (100%, 80% and 60% of evapotranspiration, ET) for different morphophysiological, yield and quality traits of cucumber. Although the results of the study could provide useful information on efficient protected structure combined with optimal utilisation of scarce water to produce high yields and high-quality cucumber fruits, there are some points that should be addressed before MS is considered for publication in Plants.

Response: We thanks reviewer for the appreciation and remarks on our research work, as well as the important suggestions provided to improve the manuscript's quality.

Comment #2: The introduction is well written! But how can the author determine the optimal amount of irrigation if he has no sensors to monitor the soil water content?

Response: Thanks for positive comment on writing. The amount of irrigation water was determined by estimating crop evapotranspiration using standard procedure; same has been described in material and methods section. Moreover, we have used soil moisture sensor (TDR) under normal level of irrigation at two stages to verify status f moisture contents before start of the subsequent irrigation the next day.

Comment #3: M&M: The description of the experimental design is poorly written. It is a two-factorial experiment with a split-plot design. Please add information on which treatment was placed in the main plot and which in the subplot.

Response: As per the valuable suggestion, we re-analyzed the data as per split plot design using R software and accordingly revised the significance to highlight the effects of the two factors on different parameters. Protected structure was placed in the main plot and irrigation levels in subplots. Accordingly, the results (significance) are revised in the tables.

Comment #4: The statistical analysis should be corrected according to the experimental design. Please include a description of the statistical methods used to statistically analyse the data.

Response: As suggested, the statistical analysis using split-plot design is corrected in the revised manuscript. Accordingly, the description of statistical method used to analyze the data is incorporated.

Comment #5: L 194: Which plunger was used to measure the firmness of the fruit??

Response: The used plunger was 6 mm and 15mm long, same is also incorporated in the revised MS.

Comment #6: Results: All results where the interaction of two factors was significant should be presented in a separate table or figure – as differences between the mean scores of the treatments with the standard errors.

Response: The result for the traits where interaction was significant (for pooled data) is incorporated as the line curve and mean differences significance assigned with standard errors and post-hoc test in the revised MS.

Comment #7: P7, 8 and 10: The nature of the figures should be corrected. Histograms are not suitable for presenting the results of the averages of the treatments. Also, in this study, the results should be presented according to the experimental design: the main plots are structures and the subplots are irrigation regimes. Please correct all figures (1, 2 and 3).

Response: Thanks for valuable suggestion. The figures for the significant interaction traits (for pooled data) are presented as line curve in the revised MS in due consideration of main plot and subplot using split-plot statistical analysis.

Comment #8: L: 388: How did the authors achieve the normal water supply?

Response: The normal water supply (crop water requirement) was estimated by considering daily (previous day’s) water loss by crop (transpiration) and soil-surface (evaporation) referred as crop evapotranspiration (ET) using standard procedures; considering daily pan evaporation reading, pan coefficient, crop factor (based on stages) to compute crop water requirement (ET). The estimated respective amount of water was precisely applied daily across the three structures with PCND (pressure compensating non-drain) inline emitter of 1 LPH.

Comment #9: If the authors include screen and nets in their considerations, they should show the restriction for quantity and quality of irradiation within the structure!

Response: The amount (quantity) of irradiation (PAR, photosynthetically active radiation) was monitored in three structures, and is presented in the materials and method section. The effect of same also has been highlighted in discussion section to relate with the microclimate of the structure (light shading/ diffusion). The quality of irradiation wasn’t monitored in this experiment, however considering your suggestion and the significance of quality of radiation, we shall include in our future research.

Comment #10: When the authors talk about the different irrigation methods, they should show the dynamics of the water content in the soil. The main drawback of the manuscript is the lack of measurements of soil water content during the growing season. The morphological and physiological response to different microclimatic conditions under the different low-tech structures and different irrigation practices could be much more accurately represented if these types of measurements were included in the study.

Response: We agree with the reviewer point on the importance of measurement of soil moisture in different treatments. Though, we had measured soil moisture at two events using time domain reflectometry (TDR) sensor borrowed from other lab to verify the moisture status in the normal irrigation level applied based on crop evapotranspiration (ET) which was found optimum as the moisture content in normal daily irrigation was maintained about 84-86% of field capacity before the start of next irrigation. Moreover, this is a standard practice to apply irrigation based on crop ET in case where sensors are not used. This method involves crop factors depending on crop stages, pan-evaporimeter reading, pan factor, and irrigation efficiency. The detailed methodology of calculating water requirement for normal irrigation and respective deficit levels are incorporated in materials and method section (2.2). The physiological parameters (e.g., RWC, WP and PS II efficiency) exhibited distinct trend with regard to irrigation levels and also due to modified environments under structures and their effects on various growth and yield parameters are clearly evident and the same are supported by observed parameters and also with related literatures.

We again thank reviewer for providing valuable suggestions. 

Sincere regards

Reviewer 2 Report

Comments and Suggestions for Authors

General comments:

The submitted article entitled “Optimization of Low-tech Protected Structure and Irrigation 2 Regime for Cucumber Production under Hot Arid Regions of 3 India” fits with the general scope of the Journal Plants MDPI. The research deals with an interesting topic which is highly relevant, especially to arid agriculture, where both harsh environment and poor water availability are prevalent, and their implication instigate difficulty in crop production in current as well as future scenario of climate change. I believe this paper will provide an insight to the researchers for finding a modest way of crop cultivation using relatively less energy and water for achieving the goal of sustainable crop production in mild to harsh climate, particularly in the era of climate change. In my opinion, the study is well planned and executed, and the findings are interesting, especially for farmers who have to cultivate under such conditions. However, I have noticed some minor flaws in different sections which need to be addressed by the authors before publication.

Abstract:

The abstract is well structured and organized, I have only minor corrections to suggest.

Line 19: Replace ‘irrigation regimes’ by ‘levels of irrigation’

Line 20: Assessed instead of evaluated

Line 21: Delete ‘these’

Line 24-28: Split sentence into two.

Line 33: replace different by the

Line 38: Delete ‘while comparing other structures’

Introduction:

the introduction provides adequate elements for framing the problem and arriving at the purpose of the work. It is well organised and provided with an adequate number of references.

Please include the scientific name of cucumber.

Line 63: Keep 'meet' in place of 'meeting'

Materials and Methods

I suggest the authors move paragraph 2.5 immediately after 2.1 so as to describe all the experimental conditions.

Line 219: the abbreviation should be uniform through MS. It is suggested to replace moderate stress (MS) to moderate deficit (MD) and severe stress (SS) with severe deficit (SD).

L227: Similarly, in table change to MD and SD in parenthesis.

Results

Results are clearly presented. Figures and tables are of high quality. Sometimes the authors use the wrong font size for the table title. Please, carefully check the guidelines of the Journal. Other minor comments below.

Line 243: typo error “two years”

Line 244: correct abbreviation for shade screenhouse “SHS” as used elsewhere

In figure 1 and also figures in the MS, mention the details about post-hoc significance test applied to compare means and what capital and small letter indicate in the bar chart.

Line 315: Use “efficiency” instead of “value” for better representation of photosystem II efficiency

Line 341: Insert comma after however

Discussion

The discussion section is complete and explains the results well, I only have a few suggestions to further enhance it.

Line 474: Insert comma after “fact”

Line 488: Delete “in” after  “of” and it should be “water supply conditions”

Line 490: Use “were clearly” instead of “was clearly”

Line 510: ‘decreases’ in place of ‘is reduced’

Line 504: insert justification of higher water productivity under NVP than other structures.

Line 520: It should be “80% of ET” instead of 800%.

References

Carefully check the references, sometimes the bold is missing (e.g. line 588) and sometimes there are errors like in line 586 (2021b)

In view of the above, authors are advised to revise the manuscript following the suggestions/ comments made by me, highlighting the changes in different colour or in track changes mode. Consequently, I consider this manuscript suitable for publication only after the MINOR changes suggested.

Author Response

Dear reviewer#2

Thanks for sparing your time and constructive suggestions to improve the quality of the manuscript. The revision has been made in the revised version in Red fonts considering comments of all the three reviewers.

General comments:

The submitted article entitled “Optimization of Low-tech Protected Structure and Irrigation  Regime for Cucumber Production under Hot Arid Regions of India” fits with the general scope of the Journal Plants MDPI. The research deals with an interesting topic which is highly relevant, especially to arid agriculture, where both harsh environment and poor water availability are prevalent, and their implication instigate difficulty in crop production in current as well as future scenario of climate change. I believe this paper will provide an insight to the researchers for finding a modest way of crop cultivation using relatively less energy and water for achieving the goal of sustainable crop production in mild to harsh climate, particularly in the era of climate change. In my opinion, the study is well planned and executed, and the findings are interesting, especially for farmers who have to cultivate under such conditions. However, I have noticed some minor flaws in different sections which need to be addressed by the authors before publication.

Response: Thanks for the appreciation on the importance and quality of the manuscript.

Abstract: The abstract is well structured and organized, I have only minor corrections to suggest.

Response: Thanks for the positive comment

Line 19: Replace ‘irrigation regimes’ by ‘levels of irrigation’

Response: Corrected made in the revised MS

Line 20: Assessed instead of evaluated

Response: Corrected

Line 21: Delete ‘these’

Response: Corrected

Line 24-28: Split sentence into two.

Response: We have made the sentence into two.

Line 33: replace different by the

Response: Corrected

Line 38: Delete ‘while comparing other structures’

Response: Corrected

Introduction: the introduction provides adequate elements for framing the problem and arriving at the purpose of the work. It is well organized and provided with an adequate number of references.

Response: Thanks for the comments

Please include the scientific name of cucumber.

Response: Included

Line 63: Keep 'meet' in place of 'meeting'

Response: Corrected

Materials and Methods

I suggest the authors move paragraph 2.5 immediately after 2.1 so as to describe all the experimental conditions.

Response: Corrected

Line 219: the abbreviation should be uniform through MS. It is suggested to replace moderate stress (MS) to moderate deficit (MD) and severe stress (SS) with severe deficit (SD).

Response: Corrected throughout the MS

L227: Similarly, in table change to MD and SD in parenthesis.

Response: Corrected

Results

Results are clearly presented. Figures and tables are of high quality. Sometimes the authors use the wrong font size for the table title. Please, carefully check the guidelines of the Journal. Other minor comments below.

Response: Thanks for the positive and valuable comment

Line 243: typo error “two years”

Response: Corrected

Line 244: correct abbreviation for shade screenhouse “SHS” as used elsewhere

Response: Corrected

In figure 1 and also figures in the MS, mention the details about post-hoc significance test applied to compare means and what capital and small letter indicate in the bar chart.

Response: Corrected

Line 315: Use “efficiency” instead of “value” for better representation of photosystem II efficiency

Response: Corrected

Line 341: Insert comma after however

Response: Corrected

Discussion

The discussion section is complete and explains the results well, I only have a few suggestions to further enhance it.

Response: Thanks for the suggestions

Line 474: Insert comma after “fact”

Response: Corrected

Line 488: Delete “in” after  “of” and it should be “water supply conditions”

Response: Corrected

Line 490: Use “were clearly” instead of “was clearly”

Response: Corrected

Line 510: ‘decreases’ in place of ‘is reduced’

Response: Corrected

Line 504: insert justification of higher water productivity under NVP than other structures.

Response: It is included

Line 520: It should be “80% of ET” instead of 800%.

Response: Corrected

References

Carefully check the references, sometimes the bold is missing (e.g. line 588) and sometimes there are errors like in line 586 (2021b)

Response: Corrected the errors

In view of the above, authors are advised to revise the manuscript following the suggestions/ comments made by me, highlighting the changes in different colour or in track changes mode. Consequently, I consider this manuscript suitable for publication only after the MINOR changes suggested.

Response: Corrected and changes are highlighted in red colour fonts.

Thanks and regards

Reviewer 3 Report

Comments and Suggestions for Authors

The mansucript is well written. I suggest the authors to edit the figures, so they are more clear (higher resolution), since now it is difficult to see the letters. Otherwise i have no other complains about the manuscript.

Author Response

Dear Reviewer#3

Thanks for sparing your time and providing useful suggestions to improve the quality of the manuscript.

The revision has been made in the revised version of manuscript in Red fonts, considering comments of all the three reviewers.

Sincere regards

Round 2

Reviewer 1 Report

Comments and Suggestions for Authors

Dear Authors,

Thank you for considering all the above suggestions for improving your article.

Author Response

Dear Reviewer,

Thanks for your positive comment on the revision.

Regards